# Comparative analysis of DNA extraction and PCR product purification methods for cervicovaginal microbiome analysis using *cpn*60 microbial profiling

Elinor Shvartsman[1,2,3☯], Meika E. I. Richmond[1,2,3☯], John J. Schellenberg[2,3☯], Alana Lamont[4,5], Catia Perciani[6], Justen N. H. Russell[2,6], Vanessa Poliquin[5], Adam Burgener[4,5,7], Walter Jaoko[8], Paul Sandstrom[2,3], Kelly S. MacDonald[1,2,3,6]*

1 Department of Internal Medicine, Max Rady College of Medicine, University of Manitoba, Winnipeg, Canada, 2 JC Wilt Infectious Diseases Research Centre, Public Health Agency of Canada, Winnipeg, Manitoba, Canada, 3 Department of Medical Microbiology and Infectious Diseases, Max Rady College of Medicine, University of Manitoba, Winnipeg, Canada, 4 Center for Global Health and Diseases, Case Western Reserve University, Cleveland, OH, United States of America, 5 Department of Obstetrics and Gynecology, University of Manitoba, Winnipeg, Canada, 6 Department of Immunology, University of Toronto, Toronto, Canada, 7 Department of Medicine, Karolinska Institute, Solna, Sweden, 8 Kenyan AIDS Vaccine Initiative–Institute of Clinical Research (KAVI-ICR), University of Nairobi, Nairobi, Kenya

☯ These authors contributed equally to this work.
* kelly.macdonald@umanitoba.ca

## Abstract

### Background

The microbiota of the lower female genital tract plays an important role in women's health. Microbial profiling using the *chaperonin*60 (*cpn*60) universal target (UT) improves resolution of vaginal species associated with negative health outcomes compared to the more commonly used 16S ribosomal DNA target. However, the choice of DNA extraction and PCR product purification methods may bias sequencing-based microbial studies and should be optimized for the sample type and molecular target used. In this study, we compared two commercial DNA extraction kits and two commercial PCR product purification kits for the microbial profiling of cervicovaginal samples using the *cpn*60 UT.

### Methods

DNA from cervicovaginal secretions and vaginal lavage samples as well as mock community standards were extracted using either the specialized QIAamp DNA Microbiome Kit, or the standard DNeasy Blood & Tissue kit with enzymatic pre-treatment for enhanced lysis of gram-positive bacteria. Extracts were PCR amplified using well-established *cpn*60 primer sets and conditions. Products were then purified using a column-based method (QIAquick PCR Purification Kit) or a gel-based PCR clean-up method using the QIAEX II Gel Extraction Kit. Purified amplicons were sequenced with the MiSeq platform using standard procedures. The overall quality of each method was evaluated by measuring DNA yield, alpha diversity, and microbial composition.

**Data Availability Statement:** Relevant data are within the paper and its Supporting Information files. Additional raw fastq. sequencing files with associated metadata have been deposited on NCBI BioProject, SRA, and BioSample and will be released on December 2nd, 2021 under the BioProject accession: PRJNA783648. The data can also be accessed through the following link: https://www.ncbi.nlm.nih.gov/bioproject/?term=PRJNA783648.

**Funding:** The study was funded by CIHR Team Grant - Research Operating THA-11960. Additional funding awards are as follows: "This study was funded by the Canadian Institutes for Health Research (grant THA-11960). KSM was funded by an Ontario HIV Treatment Network (OHTN) Senior Investigator Award and is currently supported by the University of Manitoba Department of Medicine H.E. Sellers Research Chair. ES was supported by a CIHR-Fredrick Banting and Charles Best Graduate Masters Scholarship and a University of Manitoba Faculty of Graduate Studies Top-up Award. CTP was supported by a CIHR Vanier Canada Graduate Scholarship, a Delta Kappa Gamma Society World Fellowship, and an Ontario Graduate Scholarship. The funders had no role in study design, data collection and analysis, decision to publish, or preparation of the manuscript."

**Competing interests:** The authors have declared that no competing interests exist.

## Results

DNA extracted from cervicovaginal samples using the DNeasy Blood and Tissue kit, pre-treated with lysozyme and mutanolysin, resulted in increased DNA yield, bacterial diversity, and species representation compared to the QIAamp DNA Microbiome kit. The column-based PCR product purification approach also resulted in greater average DNA yield and wider species representation compared to a gel-based clean-up method. In conclusion, this study presents a fast, effective sample preparation method for high resolution *cpn*60 based microbial profiling of cervicovaginal samples.

## Introduction

The microbial community of the lower female genital tract plays a pivotal role in women's reproductive and sexual health [1, 2]. Due to this, it has become a target for microbiome studies particularly since conventional diagnostic methods and culture-based characterization have failed to elucidate the causes of symptomatic clinical disease or fully delineate states of "altered" microbiota. Like other microbiome studies of mucosal interfaces, these studies require careful attention to the primary nucleic acid extraction method, and examination of potential biases.

Optimal vaginal microbial communities are thought to be dominated by specific lactobacilli which confer health benefits via the production of antimicrobial metabolites, such as lactic acid, hydrogen peroxide and bacteriocins [3]. Bacterial vaginosis (BV) is a commonly observed non-optimal vaginal microbial community, characterized by a lower relative abundance of lactobacilli, and increased abundance of facultative or strict anaerobes [4]. BV has been linked to increased susceptibility to sexually transmitted infections including human immunodeficiency virus (HIV) infection, as well as pregnancy complications [5–7]. Despite its clinical importance, the exact etiology of BV remains enigmatic and current treatments show limited efficacy and high recurrence rates [8, 9], hence the recent expansion in interest regarding the constitutive microbiome of the female genital tract.

*Gardnerella vaginalis* is a gram-variable pleomorphic rod bacterium originally proposed as the causative agent of the clinical entity now known as BV [10]. However, it is also ubiquitous in healthy women [11, 12]. Extensive phylogenetic diversity within this genus has been observed by sequencing of the *chaperonin*60 (*cpn*60) universal target (UT), which resolved at least four molecular subgroups with potentially distinct virulence factors [13–16]. However, these subgroups cannot be distinguished by the most commonly used amplification target for microbial profiling studies, 16S ribosomal DNA (rDNA) [17]. The *cpn*60 UT, which is usually present as a single copy gene in most bacteria, mitochondria and plastids of eukaryotes, has been successfully applied to study the role of *Gardnerella* subgroups in the vaginal mucosal milieu [17–19]. Understanding the epidemiological, immunological, and clinical correlates associated with *Gardnerella* subtypes may be vital to elucidate BV etiology and treatment strategies. However, there is a need to optimize DNA extraction and PCR product purification protocols in order to improve information yield and reproducibility of *cpn*60 microbial profiling in cervicovaginal samples.

Regardless of the amplification target, the quality of the data fundamentally depends on the primary nucleic acid extraction method. Potential biases include microbial contamination of extraction reagents, differential cell lysis, host DNA contamination, and suitability of the specific extraction method to the mucosal sample type [20]. Biases introduced during DNA

isolation may distort the measurable microbial community composition and skew relative abundances, resulting in inaccurate inferences [21, 22]. Despite the importance of the DNA extraction step, protocols are often chosen without an explicit rationale and are not formally validated. In contrast to in-house extractions, commercial kits are commonly chosen due to convenience, time efficiency, and perceived improved reproducibility. However, commercial DNA extraction kits are often not optimized to the sample type or the amplification target and thus may result in inaccurate profiles of microbial species once sequenced.

The goal of this study was to compare two commercial methods for DNA extraction and PCR product purification for the extraction and subsequent characterization of microbial DNA from cervicovaginal secretion and lavage samples by sequencing the *cpn*60 UT. The QIAamp DNA microbiome (DM) kit has been designed to deplete host DNA contamination and enhance yield of bacterial DNA, using a differential lysis technique [23]. The DNeasy Blood and Tissue (BT) kit has been used in other vaginal microbiome studies [24–26], and was modified in this study to include a sample pretreatment step using lysozyme and mutanolysin to improve DNA isolation from gram-positive bacteria [21, 24]. Extracted DNA was then used in PCR targeting the *cpn*60 UT and purified using spin columns or gel-based extraction. The quality and efficiency of each method compared was evaluated based on DNA yield, microbial diversity, and microbial composition.

## Materials and methods

### Sample collection and processing

Cervicovaginal samples in this study came from two separate clinical cohorts. For the first cohort, a 1 mL aliquot of cervicovaginal lavage (CVL) (study sample VM001) was collected from a woman visiting a colposcopy clinic at the Health Sciences Centre in Winnipeg, Manitoba, Canada as part of the Vaginal Mucosal Systems (VMS) study. Ethics certification was granted by the University of Manitoba Biomedical Research Ethics Board. Participants were over the age of 18 and provided written informed consent. To collect CVL, the physician inserted a sterile syringe containing 10 mL of phosphate buffered saline (PBS, pH 7.5) into the vagina, dispensed the PBS, and then aspirated the CVL (recovered final volume varied from 5–8 mL). For the second cohort, samples were chosen from the KAVI-VZV-001 clinical trial in Nairobi, Kenya (ClinicalTrials.gov: NCT02514018), which enrolled healthy women aged 18–50. Ethics certification for the KAVI-VZV-001 trial was granted by the Kenyatta National Hospital/University of Nairobi Ethics and Research Committee, the University of Toronto Research Ethics Board and the Kenyan Pharmacy and Poisons Board, with all participants providing written informed consent. Cervicovaginal secretions (CVS) were collected using a plastic Softcup^TM (Instead/Evofem Biosciences Inc., San Diego, CA, USA) device which was inserted into the vagina for 20 minutes, collected and processed as previously described [27]. Briefly, samples were diluted, treated with a protease inhibitor and centrifuged for 5 minutes at 1600 rpm. Supernatant was removed and the pellets (CVP) were resuspended via addition of 1 mL PBS (pH 7.5) and mixed in preparation for DNA extraction. Extractions were performed on 75 μL aliquots of either CVP or CVL, with 1–4 replicates per sample per DNA extraction method (S1 Table).

### Microbial DNA extraction methods

Two commercial kits were used to extract microbial DNA from samples (Fig 1). Eight 75 μL aliquots of ZymoBIOMICS Microbial Community D6300 Standard, (Zymo Research, Orange County, CA, USA) diluted 1:1 in PBS (pH 7.5) were used as positive extraction controls (coded as MC A-D). Two mock community standards were used undiluted (coded as MC E). 75 μL

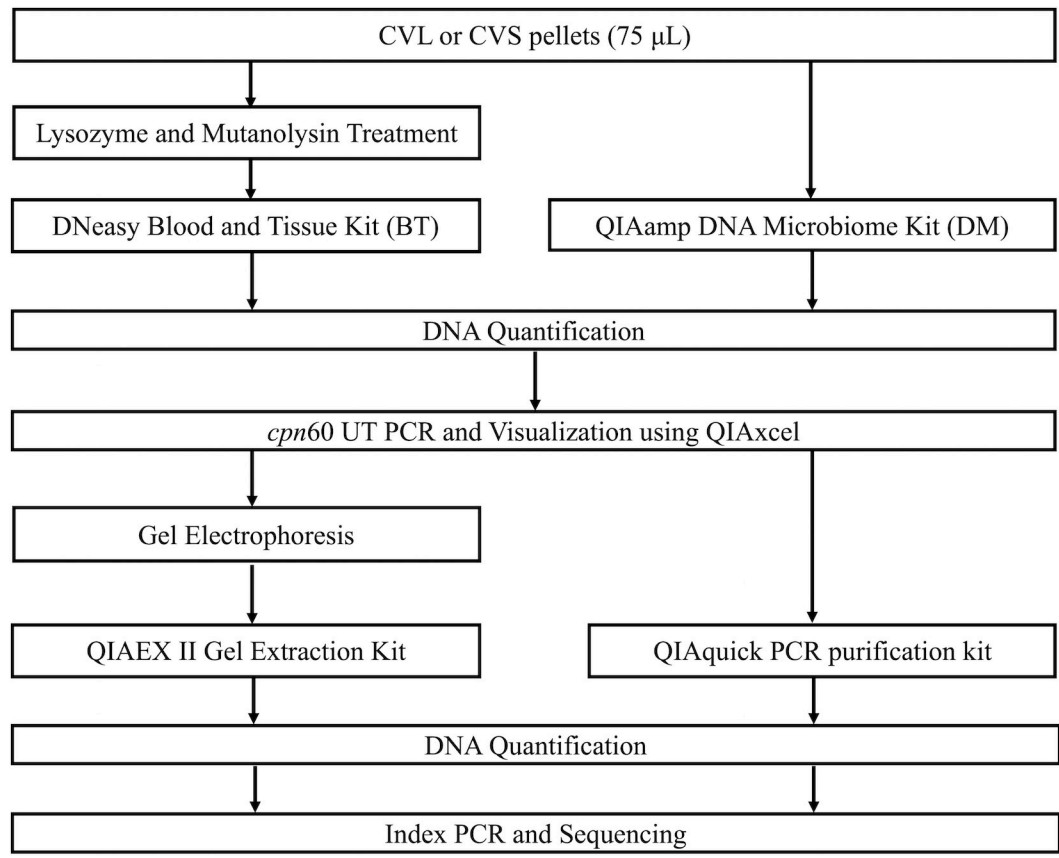

**Fig 1. Schematic representation of the DNA extraction and PCR product purification methods compared in our study.**
UT, universal target.

aliquots of PBS (pH 7.5) or microbial DNA-free water (QIAGEN Inc., Toronto, ON, Canada) were used as negative extraction controls. Extracted DNA concentrations were measured on the Qubit 2.0 fluorometer, using broad range or high sensitivity assays (Life Technologies Inc., Burlington, ON, Canada), following manufacturer's protocol. These mock community standards were developed as mixtures of six bacteria (*Pseudomonas aeruginosa*, *Escherichia coli*, *Salmonella enterica*, *Lactobacillus fermentum*, *Enterococcus faecalis*, *Staphylococcus aureus*, *Listeria monocytogenes*, *Bacillus subtilis*) and two yeasts (*Saccharomyces cerevisiae* and *Cryptococcus neoformans*). The theoretical relative abundance of each species varies considerably depending on the target for microbial profiling (genomic DNA, 16S rDNA, cell number) as described on the manufacturer's website. For *cpn*60 estimated proportions, we used the manufacturer's listed theoretical yields for genomic DNA. The inclusion of these mock community standards was done to identify potential biases introduced by the assessed extraction methods. For both extraction methods assessed in this study, DNA was eluted using 50 μL of microbial DNA-free water (QIAGEN Inc., Toronto, ON, Canada) and incubated at room temperature for 5 minutes prior to the final centrifugation step.

**DNeasy blood and tissue kit (modified).**   For the enzymatic pre-treatment, 2.5 μL of 25 U/μL mutanolysin and 10 μL of 100 mg/mL lysozyme (both from Sigma-Aldrich, Oakville, ON, Canada) were added to 90 μL of TES buffer: 10% w/v sucrose, 25 mM Tris-HCl (pH 8.0), and 10 mM EDTA. This mixture was transferred to each CVL or resuspended CVP (approximately 75 μL each) and incubated at 37˚C for 30 minutes. A 600 μL aliquot of lysis buffer (100

mM TRIS-HCl pH 8.0, 100 mM EDTA, 10 mM NaCl, 1% SDS) was added to the sample, inverted to mix, and incubated for 10–15 minutes at room temperature. Following incubation, 25 μL of proteinase K and 200 μL of buffer AL (QIAGEN Inc., Toronto, ON, Canada) were added and incubated at 56˚C for 30 minutes. Following this pre-treatment, the manufacturer's protocol for the DNeasy Blood and Tissue Kit was used (QIAGEN Inc., Toronto, ON, Canada) starting at step 3 (addition of 200 μL ethanol step).

**QIAamp DNA microbiome kit.** Manufacturer's protocol of the QIAamp DNA Microbiome Kit (QIAGEN Inc, Toronto, ON, Canada) was followed with corresponding volume adjustments for 75 μL aliquots. For step 1, a thermomixer at 600 rpm was used for the incubation. For step 6, the pathogen lysis tubes L were vortexed for 10 minutes at maximum speed.

## Polymerase chain reaction using the *cpn*60 universal target

DNA was amplified using a 1:3 ratio of *cpn*60 primer pairs (M729/M730 and M1612/M1613) modified to include MiSeq adaptors on the 5′ end (Table 1), as previously described [28, 29], with 2 μL of extracted amplicon added to 48 μL of PCR mastermix: 1X AmpliTaq Gold Buffer (Applied Biosystems, Foster City, CA), 2.5 mM MgCl$_2$, 0.2 mM dNTPs, 0.05 U/μL AmpliTaq Gold DNA Polymerase (Applied Biosystems, Foster City, CA, USA), 0.1 μM each of M729/ M730, and 0.3 μM each of M1612/M1613 primers. For mock community DNA (manufacturer-extracted), 1 μL of the ZymoBIOMICS mock community D6305 DNA standard (Zymo Research, Orange County, CA, USA) was added to 49 μL PCR mastermix as above. No template (mastermix-only) controls were also used. For PCR amplification, initial denaturation at 95˚C for 2 minutes was followed by 40 cycles of 95˚C for 30 seconds, 50˚C for 30 seconds, and 72˚C for 30 seconds, with a final extension at 72˚C for 2 minutes, as previously described [30]. Products were visualized using the QIAxcel DNA Screening Kit (QIAGEN Inc., Toronto, ON, Canada) to confirm adaptor-ligated *cpn*60 UT amplicon size (~650 bp).

## PCR product purification

Following *cpn*60 UT PCR, two amplicon purification methods were compared in replicate samples (Fig 1, S1 Table). In cases where samples failed to produce a PCR product with one or both extraction methods, that sample was omitted from the rest of the study (S1 Fig, S1 Table). In previous studies, gel electrophoresis of the *cpn*60 UT amplicons and excision of the correctly-sized band was carried out in order to minimize potential non-target amplicons [29]. Briefly, a 1% agarose gel made with 1X TAE (Tris-acetate-EDTA) buffer was loaded with a 1X rainbow dye made with equal volumes of 6X cresol red, orange-G and bromophenol blue/ xylene cyanol loading dyes (Norgen Biotek, Thorold, ON, Canada). Electrophoresis was carried out in 1X TAE buffer for 1 hour at 150V, after which the region between the cresol red and bromophenol blue (purple colored) bands (where the expected *cpn*60 product is found) was excised and extracted using the QIAEX II Gel Extraction Kit (QIAGEN, Inc., Toronto,

**Table 1. *cpn*60 PCR primers used in this investigation.**

| Name | Direction | Primer Sequence (5′-3′)[1] |
|---|---|---|
| M729 | Forward | **TCGTCGGCAGCGTCAGATGTGTATAAGAGACAG**GAIIIIGCIGGIGAYGGIACIACIAC |
| M730 | Reverse | **GTCTCGTGGGCTCGGAGATGTGTATAAGAGACAG**YKIYKITCICCRAAICCIGGIGCYTT |
| M1612 | Forward | TCGTCGGCAGCGTCAGATGTGTATAAGAGACAGGAIIIIGCIGGYGACGGYACSACSAC |
| M1613 | Reverse | **GTCTCGTGGGCTCGGAGATGTGTATAAGAGACAG**CGRCGRTCRCCGAAGCCSGGIGCCTT |

[1]Nucleotides in bold are adapters for next generation sequencing using the MiSeq platform.

ON, Canada) following manufacturer's instructions. To reduce labor and sample manipulation steps, this method was compared to a simpler spin column-based clean-up using the QIA-quick PCR purification kit (QIAGEN, Inc., Toronto, ON, Canada) following manufacturer's instructions. Purified PCR product DNA concentration was measured on the Qubit 2.0 fluorometer, using either the broad range or high sensitivity assay (Life Technologies Inc., Burlington, ON, Canada), following manufacturer's protocols.

## Sequencing and read processing

Library preparation methodology as well as downstream read processing was consistent across all samples and replicates, regardless of extraction or PCR-purification method. Sequencing adaptors were added to purified amplicon using Nextera XT Index primers (Illumina Inc., San Diego, CA, USA) and PCR was performed (two reactions per sample), according to manufacturer's directions (8 cycles of 95˚C for 30 seconds, 55˚C for 30 seconds, 72˚C for 30 seconds, followed by final extension at 72˚C for 5 minutes). Resulting amplicon libraries were purified using AMPure XP beads according to manufacturer's directions (Beckman Coulter, Mississauga, ON, Canada). DNA concentration was quantified using the Qubit 2.0 fluorometer (Life Technologies, Inc., Burlington, ON, Canada), after which the concentration of all amplicon libraries was normalized to 4 nM and pooled in preparation for loading on the MiSeq v2 500-cycle cartridge according to manufacturer's protocols (Illumina Inc., San Diego, CA, USA). The final pool of amplicon libraries was diluted to 8 pM, and 10% PhiX positive control was added and sequenced with pooled samples. Following sequencing, the.fastq file outputs were trimmed using Trimmomatic-0.36 (www.usedellab.org), with LEADING and TRAILING set to 5, required quality of read set to 15, a 4 base sliding window and MINLEN set to 120. Paired trimmed reads were aligned with *cpn*60 UT database using Bowtie2 ([http://bowtie-bio.sourceforge.net/bowtie2](http://bowtie-bio.sourceforge.net/bowtie2)), then aligned to a database derived from the VOGUE study using the mPUMA pipeline, as previously described [31, 32]. Four extraction negatives were sequenced alongside the samples (S1 File). For downstream processing of cervicovaginal samples, the average number of reads in the negative extraction controls (regardless of extraction method) for each operational taxonomic unit were removed from sample-derived reads to reduce false-positive results. For the artificial mock community analysis, *cpn*60 sequences of the eight expected species were derived from the chaperonin database ([http://www.cpndb.ca](http://www.cpndb.ca)) (S2 File). Following this, sequencing reads from the mock community standards was processed through the mPUMA pipeline with only the expected species sequences (S2 File) as possible matches.

## Statistical analysis

A two-tailed, Wilcoxon matched-pairs signed rank test was used to compare paired samples (extractions of the same sample replicate) at the alpha = 0.05 level of significance. Two diversity indices (Shannon's and "Observed" diversity) were calculated for the samples using the phyloseq package (version 1.24.2) on RStudio (version 3.5.1). For diversity and read analyses, measures from duplicate MiSeq amplicon runs were averaged for subsequent paired comparisons using the Wilcoxon matched-pairs signed rank test. Statistical analyses and figures were generated using GraphPad Prism 8.

## Results

### DNA yield following extraction and PCR product purification

Extractions were performed in replicates of two separate aliquots of the same sample (Fig 1). As expected, all negative extraction controls yielded DNA concentrations that were too low to

detect (<0.05 ng/μL). The DM kit extractions resulted in much lower DNA yield compared to the DNA extracted using the BT kit in most samples (P<0.0001, Fig 2A and 2B), with several samples below the range of detection. The total DNA extracted using the same kit was generally similar for all replicates from the same sample, although some variability was noted, particularly for the DM kit extractions (Fig 2A and 2B). All aliquots extracted with the BT kit amplified successfully and produced a strong band at approximately 650bp when visualized (S1 Fig). Most aliquots extracted with the DM kit failed to amplify and PCR product bands appeared only for a portion of the cervicovaginal samples, and mock community extraction and DNA standards (S1 Fig). All negative extraction controls failed to amplify regardless of the extraction kit used, as did the no-template controls, demonstrating little to no DNA contamination during the extraction and initial amplification procedures. Due to the failure of some of the DM aliquots to amplify, we only proceeded with PCR product purification for aliquots that successfully amplified in duplicate using both extraction methods, including eight of the aliquots and two mock community DNA standards (pre-extracted by the manufacturer). On average, spin column purification resulted in greater DNA yield compared to gel-based product purification (Fig 3A and 3B), however, this difference was only statistically significance for the cervicovaginal samples (P = 0.04) and not the mock community DNA standards. In summary, the BT kit combined with spin column purification resulted in a greater yield of amplicon for sequencing, while also being less labor-intensive than the other tested methods.

## Profiling mock community standards

We examined the sequencing output from each extraction and PCR product purification method to estimate biases of these methods in the detection of certain species in the mock communities. Specifically, manufacturer extracted mock community standards were amplified and sequenced alongside commercial mock community standards extracted with BT and DM

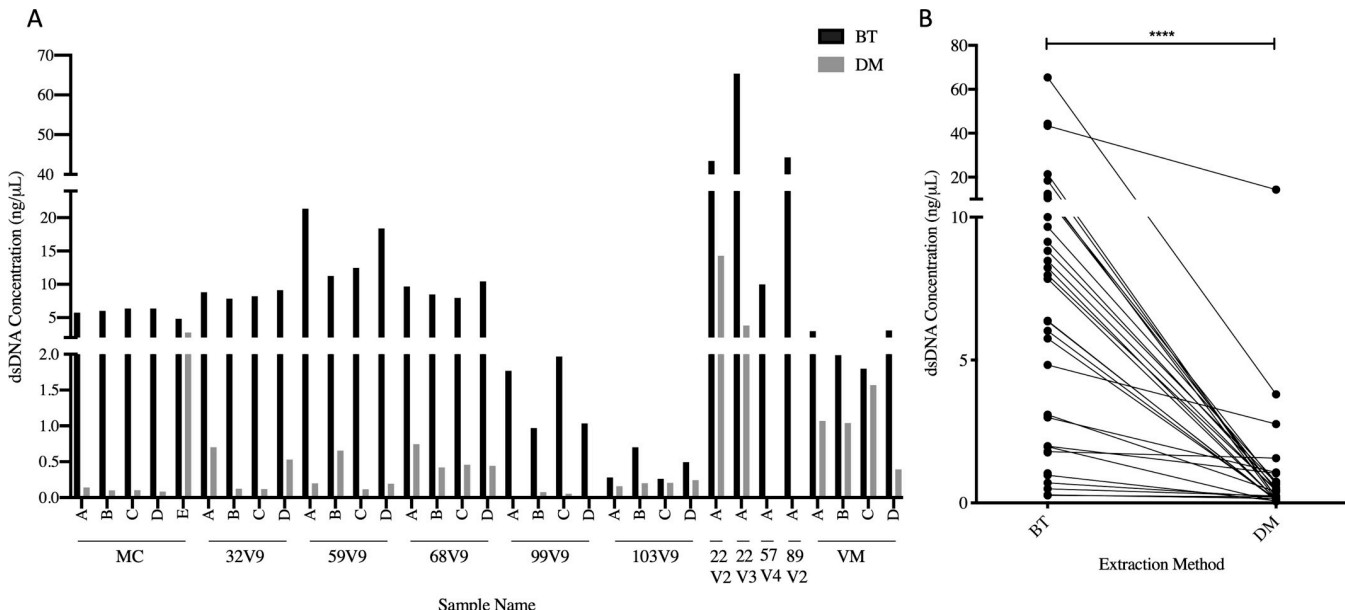

**Fig 2. Comparison of total double stranded DNA (dsDNA) yield per sample following DNA extractions as measured using the Qubit 2.0 fluorometer.** (A) Overall DNA yield per sample and replicates, (B) Paired DNA yield comparisons per extraction method used. DM, QIAamp DNA Microbiome kit, BT, modified DNeasy Blood and Tissue kit. ****P < 0.0001.

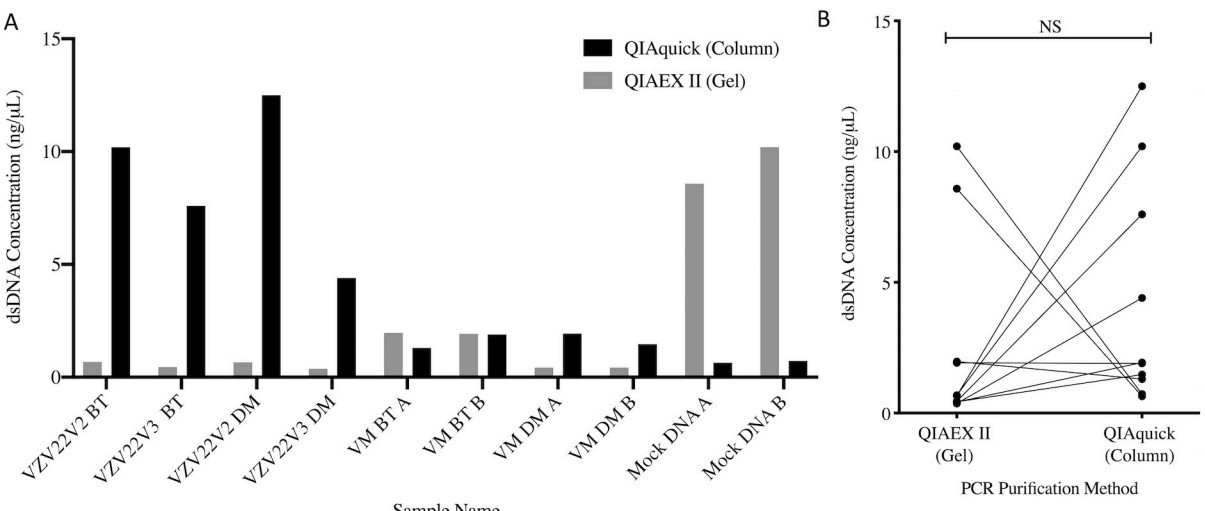

**Fig 3. Comparison of total double stranded DNA (dsDNA) yield per sample following PCR product purifications as measured using the Qubit 2.0 fluorometer using either gel-based clean-up (QIAEX II) or spin columns (QIAquick).** (A) Overall DNA yield per sample, (B) Paired comparison per purification method used. DM, QIAamp DNA Microbiome kit, BT, modified DNeasy Blood and Tissue kit protocol. NS, non-significant.

protocols, to identify biases in our extraction protocols. Neither extraction method (BT and DM) nor the sequenced manufacturer extracted standards resulted in accurate profiles of the expected mock community composition and proportion of each species. Output for the mock community DNA standards (pre-extracted by the manufacturer) captured more bacterial species than our extracted mock standards, with very similar profiles regardless of amplicon purification method (Fig 4A). DM extracted mock community standards more closely resembled the manufacturer-extracted mock community DNA standards than BT extracts (Fig 4B). Across the board, *S. enterica* was over-represented in nearly all the mock standards assessed, regardless of whether they were manufacturer pre-extracted DNA standards or extracted by us, and irrespective of the purification method. Yeast species (*S. cerevisiae* and *C. neoformans*), which were expected at an overall relative abundance of 4%, were not detected in any of the mock community standards.

Our goal was to examine sequencing output from two extraction methods and two purification methods to determine biases with respect to the detection of certain species in the mock community standards. We showed that over-representation of gram-negative bacteria and *S. enterica* in particular, as well as poor ability to capture gram positive organisms was not specific to any extraction method or purification method and also appeared in the manufacturer pre-extracted mock community standards (Fig 4A and 4B). *P. aeruginosa*, a nosocomial pathogen which has also been detected as part of the normal gut microbiota, has been underrepresented in most of our mock community standards regardless of the extraction method, although this species was over-represented in a BT-extracted mock community which had 10-100X lower sequencing depth (number of total reads) compared to the other mock controls. Mock samples extracted using the BT kit had lower sequencing depth and coverage compared to DM extracted mocks. Unlike BT extracted mock community standards, DM extracted mock community standards were able to detect *S. aureus*. *E. coli* was generally observed at the expected proportion, as was the gram-positive commensal *L. fermentum*, confirming our extraction methods can successfully capture relevant gram-positive species.

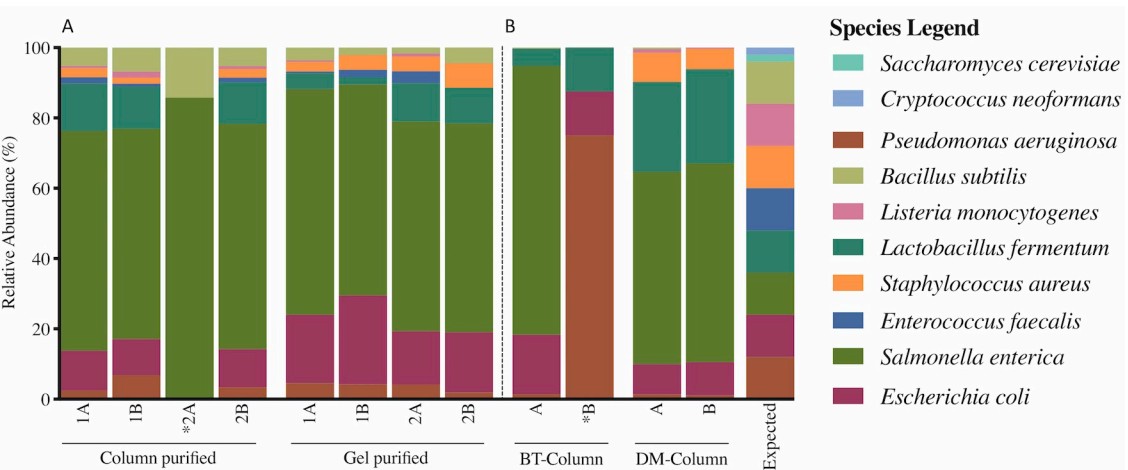

**Fig 4. Stacked bar plot comparing output of mock community standards between the different DNA extraction and PCR product purification methods.** (A) Manufacturer-extracted DNA from mock community standards, (B) Mock community standards extracted using the DM/BT kits. DM, QIAamp DNA Microbiome kit, BT, modified DNeasy Blood and Tissue kit protocol. * indicates sample had low sequencing depth (less than 20 total microbial reads). Column: refers to spin-column purification approach (QIAquick PCR purification kit). Gel: refers to gel-based PCR purification (QIAEX II kit). Aliquot numbers refer to specific extraction or PCR purification method replicates, and the follow up letters indicate a sequencing duplicate of that specific replicate.

## Profiling microbes and host coverage in cervicovaginal samples

On average, cervicovaginal BT kit extractions resulted in greater sequencing depth and species coverage compared to DM extractions, although this did not reach statistical significance (P = 0.084, Fig 5A). Analysis of observed diversity (richness, or number of species detected) and Shannon diversity (which also considers evenness, or relative distribution of species in a sample) for each extraction method, averaging output from duplicate runs, showed that BT extracted aliquots were richer (P = 0.045) and more diverse (P = 0.002) than DM kit extracted samples (Fig 5B and 5C). A very low number of eukaryotic host reads were identified in both BT and DM extracted samples, with no observed differences in host coverage (total reads attributed to host per sample) between kits (P = 0.375) (Fig 5D). Compositional output was similar across both extraction and purification methods for a given aliquot, although certain discrepancies were noted (Fig 6A–6C). Overall, the DM and BT extraction methods were in agreement in terms of the most abundant bacterial species within a sample, regardless of the PCR purification method (Fig 6A–6C). BT extractions were more likely to detect *L. iners*, *L. reuteri*, and *Oscillibacter*, a gram-negative bacterium. In contrast, the relative abundance of *L. coleohominis* was higher in DM compared to BT extracted samples. Similarly, BT extracted aliquots of the VM sample purified using spin-columns captured wider species richness and evenness compared to DM extracted and gel purified samples (Fig 6C). While the VM sample was dominated by *G. vaginalis* subgroup A, regardless of extraction or purification method used, only the BT extractions resulted in detection of *Prevotella timonensis*, *Porphyromonas uenonis* and *Dialister microaerophilus*, which are known as important BV-associated bacteria (Fig 6C).

## Discussion

DNA extraction is arguably the most important step in the characterization of the human microbiome and becomes increasingly important when species exist at lower concentrations and in complex mixtures. Optimization studies aim to improve microbial DNA yield and

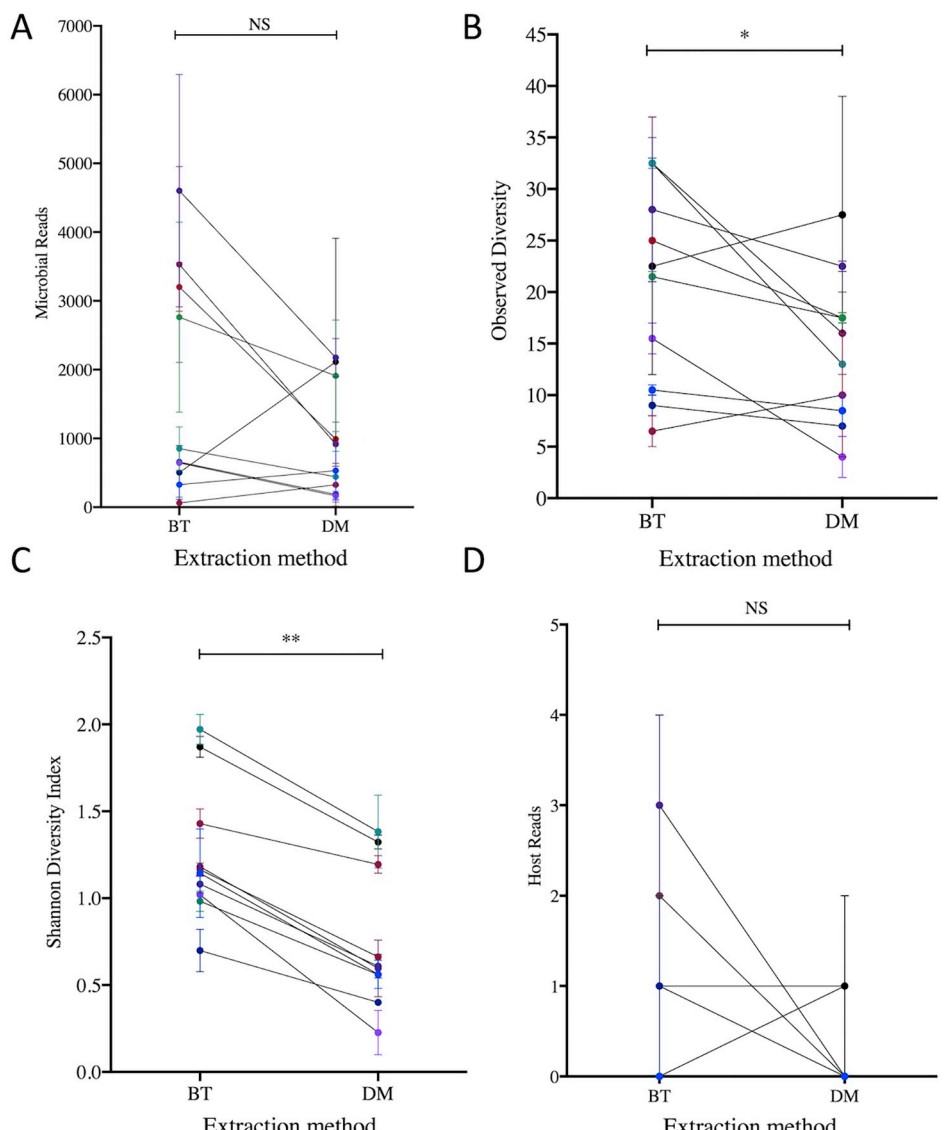

**Fig 5. Paired comparisons of sequencing read output and microbial diversity of cervicovaginal secretion and lavage samples processed using two different DNA extraction kits.** (A) Microbial read output comparison between DNA extraction methods, (B) Observed diversity representing total different microbial species compared between the two extraction methods, (C) Shannon diversity index compared between the two extraction methods, (D) Paired host (non-microbial eukaryotes) coverage comparison between DNA extraction methods. Note that the measures had been averaged per MiSeq run duplicates. Error bars represent the standard error of the mean, DM, QIAamp DNA Microbiome kit, BT, modified DNeasy Blood and Tissue kit protocol. *P<0.05, **P < 0.01, NS, non-significant.

quality, species representation and diversity, and reduce host DNA contamination. In this investigation, we compared two commonly used DNA extraction methods and two PCR product purification methods for the characterization of vaginal microbial communities using *cpn*60 microbial profiling. We compared a microbiome-specific extraction kit (DM), designed to improve lysis of bacterial cells and reduce host DNA contamination by using a differential lysis approach, to a standard DNA extraction kit used in many vaginal microbiome studies, modified to include pre-treatment with lysozyme and mutanolysin (BT). Several investigations have been conducted that compare DNA extraction methods from various anatomical

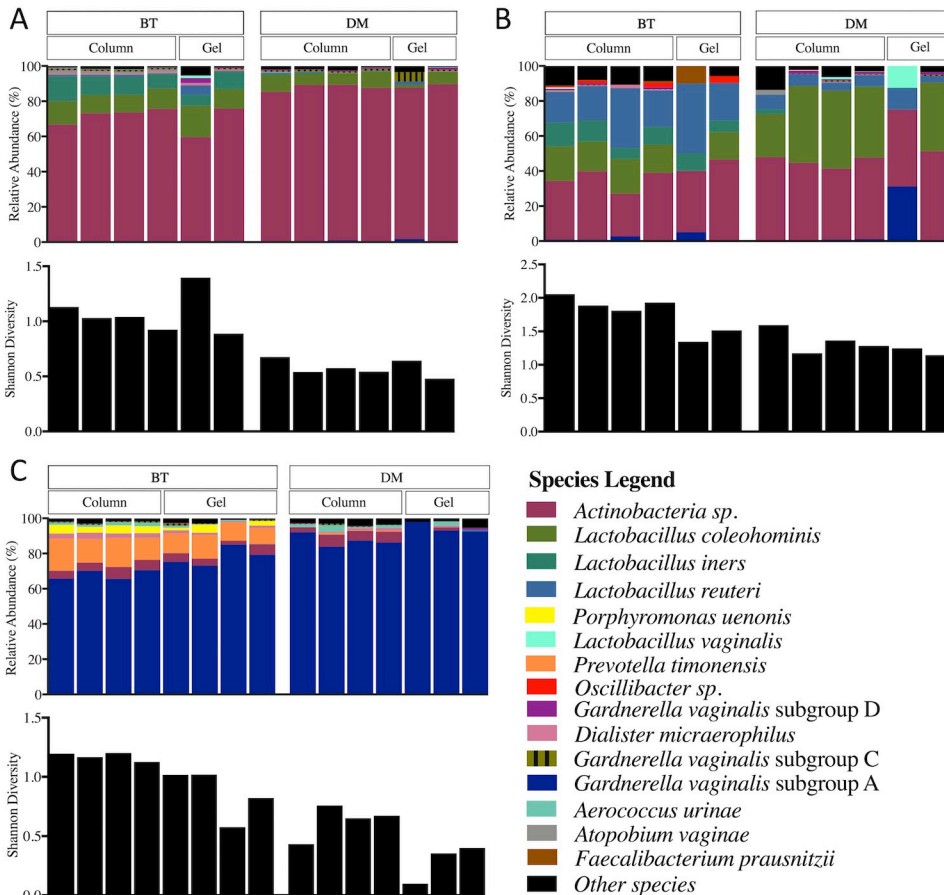

**Fig 6. Stacked bar plots representing the relative abundance of bacterial species and the corresponding Shannon diversity for vaginal samples processed using different DNA extraction and PCR product purification methods.** (A) Microbial profiles from VZV022V02 cervicovaginal pellet sample, (B) Microbial profiles from VZV022V03 cervicovaginal pellet sample, (C) Microbial profiles from the VM cervicovaginal lavage sample, DM, QIAamp DNA Microbiome kit, BT, modified DNeasy blood and tissue kit protocol. Only the 15 most prevalent species based on average relative abundance across samples are individually shown, with the remaining binned into the "other species" category.

compartments and mock communities using 16S rDNA or whole genome sequencing approaches. However, extraction methods should be optimized and validated for sample type and profiling method (here used the *cpn*60 UT), due to differences in chemical composition and ratio of microbial biomass to host-derived biomass, that can impact extraction efficiency and downstream analysis [20, 21, 33–36].

The DNA extraction methods compared in this study resulted in significant and perhaps unexpected differences in DNA yield, as well as notable differences in the characterization of bacterial species diversity and composition. Unlike the DM kit, which has been tailored to extract microbial DNA, the modified BT kit protocol with enzymatic lysis of the bacterial cell wall, yielded DNA above the limit of detection (>0.05 ng/μL) for all aliquots and replicates. In addition, amplification of the extracted DNA using a standard *cpn*60 protocol resulted in many of the DM kit samples failing to amplify, greatly reducing our downstream sample size, whereas all the BT kit samples successfully amplified. The DM kit is claimed by the manufacturer to be specific for viable microbial DNA, while the BT kit is advertised as a non-specific DNA extraction protocol that extracts DNA from both host and microbes, possibly explaining

differences in DNA yield and amplification between the two methods. Our results are concordant with a recent investigation of vaginal swab samples using 16S rDNA microbial profiling [25], which showed that the BT kit protocol (albeit without the enzymatic pre-treatment step) provided the highest overall DNA yield and quality compared to other modified commercial DNA extraction protocols, especially modified Mo Bio Laboratories PowerSoil kits.

The inclusion of synthetic mock community standards is useful in microbiome studies as it allows identification of the source of potential biases [37, 38]. In this study, microbial relative abundance did not confirm expected values from commercially-supplied mixtures of bacterial/fungal cells at known concentrations), regardless of extraction or PCR product purification methods used. The DM kit appears to provide a closer representation of the species observed for manufacturer-extracted mock communities as compared to the BT kit, although further work will be required to formally evaluate this potential advantage. Despite apparent amplification and/or extraction/PCR product purification biases, both gram-positive and gram-negative species were observed as expected. Well-characterized shotgun sequenced metagenomes of common vaginal isolates from clinical material may provide better positive controls for this type of study in the future.

In the context of the vaginal microbiome, where diverse communities including those linked to BV and aerobic vaginitis have been linked to negative health outcomes [5–7, 39], understanding how DNA extraction methods affect vaginal microbial patterns is imperative. When comparing extraction methods for cervicovaginal samples, the BT extractions resulted in higher species diversity and indicated the presence of gram-negative bacteria associated with BV, including *P. timonensis*, *P. uenonis* and *D. microaerophilus* [40]. The BT kit also indicated higher relative abundance of *L. iners* in one of the samples. *L. iners* has been associated with vaginal microbial shifts and transitional states such as those occurring following BV treatment, and thus may play a crucial role in understanding vaginal microbial dynamics [41, 42]. Triangulation of these results with metagenomic profiles or quantitative PCR would help to determine which method more accurately represents the actual distribution of *L. iners* and other taxa of interest. Indices of microbial community diversity (within-sample species richness and evenness) suggest that BT extractions provide more in-depth information due to higher within-sample diversity when compared to DM extractions.

Studies using 16S rDNA as the phylogenetic target naturally exclude host DNA "contamination" since it is exclusively found in prokaryotes. However, the *cpn*60 UT is found in both prokaryotes and eukaryotes, allowing for analysis of potential host *cpn*60 coverage in the sequencing output. The ratio of microbial biomass to host-derived cells in a sample volume is critical since information about microbial communities may potentially be missed due to repetitive host signal. The DM kit was designed to preferentially isolate microbial DNA in the context of high host background (for example in blood, cerebrospinal fluid or bronchoalveolar lavage fluid). In this study, overall host coverage was low in vaginal specimens, and did not differ significantly between the two extraction methods used. However, we cannot attest to the true host DNA contamination in our study as we did not assess the true proportion of the DNA yield that is exclusively microbial, for example by quantitative PCR. Further studies will be required to examine the true discrepancy in host DNA yields when using different extraction methods.

PCR product purification procedures can also affect sequencing output. Most standard PCR product clean-up kits based on spin columns remove primers, reagents, and other impurities which may interfere with downstream applications. Since non-specific amplification using *cpn*60 UT-targeted primers has been observed, gel purification methods can be used to specifically isolate only amplicons of the desired size range [29, 30]. In this study, only strong bands of the correct size were observed by electrophoresis, indicating that size-based

purification was unnecessary and resulted in significantly lower average DNA yields compared to spin columns. Microbial diversity and species composition were also higher when using column purification, however our very small sample size hampers our ability to detect statistically significant differences.

To conclude, our findings suggest that using the non-microbiome specific kit (BT) with an added enzymatic pre-treatment results on average, in higher DNA yield, bacterial diversity, and representativeness compared to the more labor-intensive microbiome-specific DNA extraction kit (DM), with both methods showing similar low host coverage. When using *cpn*60 microbial profiling to study the cervicovaginal microbiome with no increase in host reads we also show that purification of PCR products using a column-based approach results in relatively higher yield of species richness than a gel-based PCR clean up method. Aside from advanced metagenomics and metaproteomic analyses, *cpn*60 UT-based profiling of the cervicovaginal microbiome remains one of the only methods to distinguish subtypes of *Gardnerella*.

## Supporting information

**S1 Table. Study sample catalogue outlining the number of cervicovaginal samples and mock community standards included in the study, alongside the number of replicates per extraction and PCR product purification method.**
(XLSX)

**S1 Fig. Capillary electrophoresis of the vaginal samples and mock communities done on two runs to identify *cpn*60 UT product.** DM or MB refer to QIAamp DNA Microbiome kit, BT, modified DNeasy Blood and Tissue kit protocol. Numbers refer to VZV participant ID unless otherwise specified. Note that some samples were amplified alongside our study samples using *cpn*60 for another unrelated study and are also included in these raw images (for ex: cont DNA BT).
(PDF)

**S1 File. Raw sequencing read counts for the samples included in the study, and the negative extraction controls.**
(XLSX)

**S2 File. Chaperonin60 sequences used for targeted search in the analysis of the Mock community standards.**
(TXT)

## Acknowledgments

This work could not have been done without the KAVI-VZV-001 and VM study participants and the VM team, in particular Fran Mulhall, Kenzie Birse and colleagues at the Women's Health Research Institute, and the KAVI-ICR Team members (listed below). We would also like to thank Dr. Janet Hill for sharing of expertise with read processing using the mPUMA pipeline, Rupert Capina and Dr. Emma Lee for providing guidance using the sequencing platform, and the National Microbiology Laboratory DNA Core facility for assistance with normalization of DNA in preparation for sequencing.

**KAVI-ICR Team members: Investigators:** Dr. Omu Anzala. **Community-Clinic:** Roselyne Malogo, Rose Mahira, Dr. Gaudensia Mutua, Dr. Lydia Atambo, Dr. Borna Nyaoke, Jacquelyn Nyange, Judith Omungo, Timothy Kotikot, Mary W. Gichuho, Hilda Ogutu, Rose Ndambuki, Emmanuel Museve, Hannah Nduta Gakure, Dorothy Essendi, Elizabeth Mutiska.

**Laboratory:** Bashir Farah, Brian Onsembe, Matrona Akiso, Simon Ogola, Nelly Wanjiku, Robert Langat, Jackton Indangasi, Naomi Mwakisha, Irene Mwangi, Marion Agwaya, Ruth Chirchir, Richard Alila, Lewa Said. **Pharmacy:** James Wakonyo, Mercy Musanga, Catherine Kamau. **IT/Data:** Moses Muriuki, Jason Ndalamia, Catherine Ngeli, and Laura Lusike.

## Author Contributions

**Conceptualization:** Meika E. I. Richmond, John J. Schellenberg, Kelly S. MacDonald.

**Data curation:** Elinor Shvartsman, Meika E. I. Richmond, John J. Schellenberg, Alana Lamont.

**Formal analysis:** Elinor Shvartsman, Meika E. I. Richmond, John J. Schellenberg, Alana Lamont.

**Funding acquisition:** Adam Burgener, Walter Jaoko, Paul Sandstrom, Kelly S. MacDonald.

**Investigation:** Elinor Shvartsman, Meika E. I. Richmond, John J. Schellenberg, Alana Lamont, Catia Perciani, Justen N. H. Russell, Vanessa Poliquin, Adam Burgener, Kelly S. MacDonald.

**Methodology:** Meika E. I. Richmond, John J. Schellenberg, Catia Perciani, Justen N. H. Russell, Kelly S. MacDonald.

**Project administration:** Catia Perciani, Vanessa Poliquin, Walter Jaoko, Paul Sandstrom, Kelly S. MacDonald.

**Resources:** Vanessa Poliquin, Adam Burgener, Walter Jaoko, Paul Sandstrom, Kelly S. MacDonald.

**Supervision:** Adam Burgener, Walter Jaoko, Paul Sandstrom, Kelly S. MacDonald.

**Visualization:** Elinor Shvartsman.

**Writing – original draft:** Elinor Shvartsman, Meika E. I. Richmond.

**Writing – review & editing:** Elinor Shvartsman, John J. Schellenberg, Alana Lamont, Vanessa Poliquin, Adam Burgener, Paul Sandstrom, Kelly S. MacDonald.

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
