## [Decision Letter · Decision Letter 0]

18 Oct 2021

PONE-D-21-25280Comparative analysis of DNA extraction and PCR product purification methods for cervicovaginal microbiome analysis using cpn60 microbial profilingPLOS ONE

Dear Dr. MacDonald,

Thank you for submitting your manuscript to PLOS ONE. After careful consideration, we feel that it has merit but does not fully meet PLOS ONE’s publication criteria as it currently stands. Therefore, we invite you to submit a revised version of the manuscript that addresses the points raised during the review process.

We look forward to receiving your revised manuscript.

Kind regards,

Guangming Zhong

Academic Editor

PLOS ONE

Journal Requirements:

Reviewers' comments:

Reviewer's Responses to Questions

**Comments to the Author**

1. Is the manuscript technically sound, and do the data support the conclusions?

Reviewer #1: Yes

Reviewer #2: No

2. Has the statistical analysis been performed appropriately and rigorously? 

Reviewer #1: Yes

Reviewer #2: No

3. Have the authors made all data underlying the findings in their manuscript fully available?

Reviewer #1: Yes

Reviewer #2: No

4. Is the manuscript presented in an intelligible fashion and written in standard English?

Reviewer #1: Yes

Reviewer #2: No

5. Review Comments to the Author

Reviewer #1: This manuscript is written very well. The design is studied well. I do think comparing two different sets of experimental methods confounds the results a bit. Perhaps it would have been better to compare microbial yield using the two different DNA extraction kits and then only using either the column-based method or the gel-based method, instead of also comparing that second element. Inclusion of the mock community standard was a good thought.

Reviewer #2: 1) I have to say the manuscript is very unclear. I cannot even tell how many samples were involved and how many replicates were done. Across figures, why the samples are changed? There should be a good catalogue on what samples underwent what experiments.

2) Sequencing depth can be related with sequencing library preparation. How can it be accounted for before comparing the sequencing results, which is related with sequencing depth, between extraction and purification methods?

6. PLOS authors have the option to publish the peer review history of their article (what does this mean?). If published, this will include your full peer review and any attached files.

Reviewer #1: No

Reviewer #2: No

---

## [Author Response · Author response to Decision Letter 0]

1 Dec 2021

Response to reviewer #1:

Reviewer #1: This manuscript is written very well. The design is studied well. I do think comparing two different sets of experimental methods confounds the results a bit. Perhaps it would have been better to compare microbial yield using the two different DNA extraction kits and then only using either the column-based method or the gel-based method, instead of also comparing that second element. Inclusion of the mock community standard was a good thought.

We thank the reviewer for their kind words and agree that the added method comparison did add an additional layer of complexity with respect to the analysis. This has been addressed within the manuscript by addition of replicates for both extraction and PCR-clean up methods, and has been addressed during statistical analysis, where paired comparisons were performed where applicable. 

Response to reviewer #2: 

Reviewer #2: 1) I have to say the manuscript is very unclear. I cannot even tell how many samples were involved and how many replicates were done. Across figures, why the samples are changed? There should be a good catalogue on what samples underwent what experiments.

We appreciate the reviewer’s comments and think the inclusion of a sample catalogue and additional clarifications within the text have made the paper better organized. A sample catalogue has been included as a supplementary table to address questions regarding replicate number and status for the biological samples and mock standards (S1 Table) per extraction method and PCR-purification method. Footnotes have been added in the table where applicable. 

We acknowledge that there was variation in the number of replicates per sample. This was due to different persons performing the extractions and has been addressed where applicable in the figures and in the statistical analysis where error bars have been included where applicable. 

Some samples were indeed omitted from the PCR-product clean-up and the final sequencing round due to failure to amplify with one or both kits – this has been addressed in S1 Table, shown in S1 Fig, and stated within the text on lines 195-196 in the methods section and on lines 261-264 in the results section. Since the purpose of the study was to compare two extraction methods, failure to amplify with one kit resulted in omission of the sample from any downstream processes (ex: PCR clean-up, and sequencing). 

2) Sequencing depth can be related with sequencing library preparation. How can it be accounted for before comparing the sequencing results, which is related with sequencing depth, between extraction and purification methods?

We agree with the reviewer that sequencing depth (or the total number of read output per sample) is indeed related to library preparation. In our study, library preparation was consistent across all sequenced samples (same cpn60 PCR protocol was used, same index PCR, post-index PCR, etc). This has been further clarified on lines 213-214 in the text.

---

## [Editor Report · Decision Letter 1]

22 Dec 2021

Comparative analysis of DNA extraction and PCR product purification methods for cervicovaginal microbiome analysis using cpn60 microbial profiling

PONE-D-21-25280R1

Dear Dr. MacDonald,

We’re pleased to inform you that your manuscript has been judged scientifically suitable for publication and will be formally accepted for publication once it meets all outstanding technical requirements.

Kind regards,

Guangming Zhong

Academic Editor

PLOS ONE

Additional Editor Comments (optional):

The authors have adequately addressed all concerns.
---

## [Editor Report · Acceptance letter]

5 Jan 2022

PONE-D-21-25280R1 

Comparative analysis of DNA extraction and PCR product purification methods for cervicovaginal microbiome analysis using *cpn*60 microbial profiling 

Dear Dr. MacDonald:

I'm pleased to inform you that your manuscript has been deemed suitable for publication in PLOS ONE. Congratulations! Your manuscript is now with our production department. 

Kind regards, 

on behalf of

Dr. Guangming Zhong 

Academic Editor

PLOS ONE